# Serotyping and Antibiotic Resistance Profiles of *Salmonella* spp. and *Listeria monocytogenes* Strains Isolated from Pet Food and Feed Samples: A One Health Perspective [note 1]

**DOI:** 10.3390/vetsci12090844

**Published:** 2025-09-01

**Authors:** Nikolaos D. Andritsos, Antonia Mataragka, Nikolaos Tzimotoudis, Anastasia-Spyridoula Chatzopoulou, Maria Kotsikori, John Ikonomopoulos

**Affiliations:** 1Department of Food Science and Technology, School of Agricultural Sciences, University of Patras, 2 G. Seferi Str., GR-30100 Agrinio, Greece; amataragka@upatras.gr; 2Laboratory of Anatomy and Physiology of Farm Animals, Department of Animal Science, School of Animal Biosciences, Agricultural University of Athens, 78 Iera Odos Str., GR-11855 Athens, Greeceikonomop@aua.gr (J.I.); 3Hellenic Army Biological Research Centre, 6 Taxiarchou Velliou Str., P. Penteli, GR-15236 Attica, Greece; n.p.tzimotoudis@army.gr

**Keywords:** antibiotic, antimicrobial resistance, feed, *Listeria monocytogenes*, One Health, pet food, serotype, serovar, *Salmonella*

## Abstract

*Salmonella* and *Listeria monocytogenes* are two ubiquitous bacteria, pathogenic to animals and humans alike, that can be found in food and animal feeding stuff. Furthermore, antimicrobial resistance (AMR) is a growing global issue of great concern with serious public health implications. This work addresses the problem of detecting different serological variants of the above pathogens in pet food and feed, while also screening for antibiotic-resistant phenotypes among the microbial isolates. Three clinically important *Salmonella* serovars were identified in salmonellae strains recovered from pet food and feed samples (*S.* Enteritidis, *S.* Typhimurium, *S.* Thompson), whereas *L. monocytogenes* isolates were assigned to the four most prevalent serotypes of the pathogen encountered in foods (1/2a, 1/2b, 1/2c, 4b). Screening for AMR revealed the resistance to three out of nine tested antibiotics (33.3%) for *Salmonella* spp. (ampicillin, tetracycline, trimethoprim-sulfamethoxazole) and five out of seven antibiotics (71.4%) for *L. monocytogenes* (ciprofloxacin, meropenem, penicillin, tetracycline, trimethoprim-sulfamethoxazole). No multidrug resistance (MDR, i.e., resistance to at least three antimicrobial classes) was recorded in strains of *Salmonella*, but one strain of *L. monocytogenes* isolated from pet food was classified as an MDR strain. The results of the present study demonstrate the presence of AMR foodborne pathogenic bacteria in animal feeding stuffs, highlighting the potential of detecting antibiotic-resistant and MDR bacterial pathogens, such as *Salmonella* spp. and *L. monocytogenes*, in foods.

## 1. Introduction

The European Food Safety Authority (EFSA) and the European Centre for Disease Control and Prevention (ECDC) publish each year the European Union (EU) summary report on trends and sources of zoonoses, zoonotic agents, and foodborne outbreaks. In this EU report, the term “One Health” has been included in its title from 2019 and onwards [1]. Therefore, the annual report can be found today under the ‘EU One Health zoonoses report’ title. According to the World Health Organization (WHO), One Health is an integrated, unifying approach that aims to sustainably balance and optimize the health of people, animals, and ecosystems [2]. It recognizes that the health of humans, domestic and wild animals, plants, and the wider environment (including ecosystems) are closely linked and interdependent (Figure 1). The One Health approach addresses the interconnection between people, animal, plants, and their shared environment [3].

Nowadays, the concept of One Health is attracting more and more the attention of scientists and statutory and regulatory authorities, since the transition of environmental contaminants and pollutants as well as human and animal pathogens from one circle to the other (Figure 1) is common and its effect has been widely proved (e.g., COVID-19 and other pandemics) [4]. Besides, in this concept, animal feed is considered as food and many human infections have been caused due to the exposure of people to microbial pathogens via handling contaminated pet food and pet treats [5] or through the consumption of contaminated raw food of animal origin (e.g., milk and meat).

Thus, in the current context of One Health, antimicrobial resistance (AMR) is also gaining increasing attention worldwide, as it poses significant threat to public health. The development of AMR in the primary animal food production environment is primarily related to the misuse of antibiotics during treatment of zoonoses [6,7]. Misuse of antibiotics may refer to (i) over-prescription of antibiotics (i.e., high dose/increased intake of antibiotic), (ii) not finishing the entire antibiotic scheme (i.e., low dose/decreased intake of antibiotic), (iii) overuse of antibiotics in livestock, and (iv) failure to comply with antibiotic treatment regimen and waiting times (withdrawal periods) for animals under antibiotic treatment. Whatever the case may be, microorganisms may end up developing resistance to the prescribed antibiotics, rendering them ineffective for treatment of animal disease.

Among other microorganisms, pathogenic bacteria can be found in the primary animal food production environment and feed as well. Increased AMR of bacterial pathogens in food animals can lead to human infections through the consumption of food with antibiotic-resistant microorganisms, which may confer diseases difficult to treat [8]. Moreover, increased AMR can be the result of cross-resistance developed in the food processing environment for some pathogens, when cells of the microorganisms are exposed to sublethal concentrations of residual disinfectants [9,10,11,12,13]. The most common microorganisms with documented AMR in the EU include *Escherichia coli*, *Campylobacter jejuni*, *C. coli*, *Salmonella* spp., and *Staphylococcus aureus* [6,8,14]. Foodborne pathogenic bacteria, like *Salmonella* spp. and *Listeria monocytogenes*, can be detected in the primary animal food production environment (e.g., breeding and dairy farms, slaughterhouses) and pet foods and feed [15,16,17,18,19,20,21,22]. Despite being a well-reputed biological hazard for its risk of infection, with a recorded case–fatality rate of circa 20% in humans [23], interestingly, *L. monocytogenes* is not included in the EU’s yearly monitoring on AMR in bacteria from humans, animals, and food [6].

Taking all the above into account, the aim of this work was to detect and identify serological variants for two of the most common foodborne pathogenic bacteria, namely *Salmonella* spp. and *L. monocytogenes*, in pet foods and feed and then to screen for AMR of these pathogens against a panel of selected antibiotics. To the best of our knowledge, this is the first attempt to systematically document the contamination status of *Salmonella* spp. and *L. monocytogenes* in pet foods and feed in Greece and highlight the serotypes and AMR profiles of the isolates.

## 2. Materials and Methods

### 2.1. Sampling, Pathogen Isolation, and Confirmation of the Isolates

Samples of commercial pet food (BARF, i.e., biologically appropriate raw food for dogs and cats), animal feed (fodder in the form of compressed and pelleted feeds, e.g., poultry feed), and a raw feed ingredient (chicken) were collected from January 2015 until May 2022 and microbiologically analyzed for the detection of *Salmonella* spp. and *L. monocytogenes* (Appendix A). Sampling and microbiological analysis were conducted by Eurofins Athens Analysis Laboratories and the Hellenic Army Biological Research Centre. ISO 6579 and ISO 11290-1 protocols were used for the detection of *Salmonella* spp. [24,25] and *L. monocytogenes* [26] from the samples, respectively, as described elsewhere [27]. Briefly, *Salmonella* detection included the overnight pre-enrichment and final enrichment of the initial suspension of the sample in buffered peptone water (Merck, Darmstadt, Germany) and Rappaport-Vassiliadis broth (Merck) at 30 °C and 41.5 °C, respectively, followed by streaking in duplicate onto xylose lysine deoxycholate (XLD) agar (Merck). Detection of *L. monocytogenes* was carried out after primary and secondary enrichment in half-Fraser (BioKar Diagnostics, Pantin, France) and full concentration Fraser broth (BioKar Diagnosticks) at 30 °C for 24 h and at 37 °C for 48 h, respectively, followed by streaking in duplicate onto COMPASS^®^
*Listeria* agar (Biokar Diagnostics). Suspect colonies of *Salmonella* on xylose lysine deoxycholate (XLD) agar (Merck) (i.e., black colonies with reddish transparent zone around them) were subcultured on nutrient agar (LabM, Lancashire, UK) for biochemical and serological confirmation of the isolates. Biochemical testing of the isolates took place by performing utilization of triple sugar iron (TSI) agar (Merck), urea agar (Merck), and L-lysine decarboxylation medium (LDC; Merck) [24,25], while it was also achieved by assigning isolates to the genus *Salmonella* using in parallel the RapID^™^ ONE System (Remel Inc., Thermo Fisher Scientific, San Diego, CA, USA) [28]. Isolates showing typical biochemical reactions for *Salmonella* spp. were subsequently serologically tested with a polyvalent slide agglutination test comprising of poly A-S+Vi and poly H antisera (Statens Serum Institute, Copenhagen, Denmark), for the presence of *Salmonella* O-antigen (somatic), H-antigen (flagellar), and Vi-antigen (capsular). Strains that were confirmed as *Salmonella* spp., showing typical biochemical reactions and positive/negative serological reactions to all or some of the antigens [25], were further typed to serovar level.

Colonies of presumptive *L. monocytogenes* on ALOA-type media (i.e., media formulated and performing like agar *Listeria* according to Ottaviani and Agosti), such as COMPASS^®^
*Listeria* agar (Biokar Diagnostics) used in the study, were confirmed through biochemical testing of the isolates as previously described by Andritsos & Mataragas [29]. Strains that were confirmed as *L. monocytogenes* were further characterized to serogroup level [30,31].

### 2.2. Antimicrobial Susceptibility Testing (AST) and Antibiotic Resistance Profiles of Salmonella spp. and L. monocytogenes Strains

The antimicrobial susceptibility testing (AST) and antibiotic resistance of *Salmonella* spp. and *L. monocytogenes* pet food and feed isolates, along with the determination of minimum inhibitory concentration (MIC) for those antibiotics inducing resistance to any of the pathogens, were conducted using the Kirby–Bauer disk diffusion method for AST [32] and the E-test method for MIC determination to each specific antimicrobial agent [33], following the guidelines and criteria of the European Committee on Antimicrobial Susceptibility Testing (EUCAST v.15.0) [34]. The experimental procedure being followed was previously described in detail for *L. monocytogenes* [29] and was the same also for strains of *Salmonella* spp., with the exception of using Mueller–Hinton agar plates (MH; Oxoid, Basingstoke, UK) instead of MH agar plates supplemented with 5% defibrinated horse blood and 20 mg/L β-NAD (MH-F; Bioprepare, Keratea, Attica, Greece) that were used for AST of *L. monocytogenes*.

AST for bacterial pathogens comprised the following antibiotic disks supplied by Oxoid (Basingstoke): *Salmonella* strains were screened against amoxycillin-clavulanate (AMC; 30 μg), ampicillin (AMP; 10 μg), cefotaxime (CTX; 5 μg), cefoxitin (FOX; 30 μg), ceftazidime (CAZ; 10 μg), ciprofloxacin (CIP; 5 μg), gentamicin (CN; 10 μg), tetracycline (TE; 30 μg), and trimethoprim-sulfamethoxazole 1:19 (SXT; 25 μg), while AST for *L. monocytogenes* strains included screening against AMP (2 μg), benzylpenicillin (P; 1U), CIP (5 μg), erythromycin (E; 15 μg), meropenem (MEM; 10 μg), SXT (25 μg), and TE (30 μg).

### 2.3. Serotyping of Salmonella spp. and L. monocytogenes Isolates

DNA was extracted from the bacterial isolates by using a commercially available kit (Nucleospin^®^ Tissue, Macheray-Nagel GmbH & Co. KG, Düren, Germany), as per the manufacturer’s instructions and as previously reported [35]. Serotyping of *Salmonella* spp. and *L. monocytogenes* isolates was achieved through multiplex polymerase chain reaction (mPCR). More specifically, serotyping of *Salmonella* strains was performed according to Shimizu et al. [36] for the detection of *Salmonella* serovars Enteritidis, Infantis, Typhimurium, and Thompson (Table 1A). Serotypes of *L. monocytogenes* were determined by applying the well-established protocol of Doumith et al. [30] for the identification and differentiation of the pathogen’s major serotypes 1/2a, 1/2b, 1/2c, and 4b in foods (Table 1B).

## 3. Results

### 3.1. Salmonella spp. and L. monocytogenes Isolates from Pet Food and Feed Samples

Twenty-four bacterial strains of *Salmonella* spp. (15 isolates) and *L. monocytogenes* (9 isolates), each isolate recovered from different sample of pet food, animal feed, or raw feed ingredient, were isolated from pet food and feed samples. Biochemical and/or serological testing of the isolates (Appendix A) confirmed the presence of the aforementioned pathogens in the samples.

### 3.2. AST of Microbial Isolates

AST of the microbial isolates revealed that the majority of *Salmonella* strains exhibited resistance to TE (53.3%; 8/15 strains). Resistance was also recorded for *Salmonella* against AMP (13.3%) and SXT (20%) (Table 2A). All *Salmonella* isolates were found to be susceptible (100%) to AMC, CAZ, CIP, CN, CTX, and FOX, with susceptibility ratings for the remaining antibiotics (AMP, SXT, TE) ranging from 46.7% to 86.7% (Table 2A).

Regarding AST results for the nine *L. monocytogenes* isolates, the highest resistance was observed against SXT (55.6%; 5/9 strains). Resistance to other tested antibiotics ranged from 11.1% (MEM, P, TE) to 22.2% and 77.8% (i.e., strong and intermediate resistance to CIP, respectively) (Table 2B). All *L. monocytogenes* isolates were susceptible (100%) to AMP and E. Susceptibility to the remaining antibiotics varied, with percentages ranging from 0% for CIP to 88.9% for MEM, P, and TE (Table 2B).

### 3.3. Antibiotic Resistance Profiles of the Pet Food and Feed Bacterial Isolates

Out of the 15 *Salmonella* isolates, 8 exhibited resistances to at least one antibiotic (53.3%), with TE showing the highest resistance rate in *Salmonella* spp., as all 8 isolates were resistant to this particular antimicrobial (Figure 2a). Similarly, among the nine *L. monocytogenes* isolates, five exhibited strong resistance to at least one antibiotic (55.6%), with those five *L. monocytogenes* strains showing resistance against SXT (Figure 2b).

Regarding the appearance of AMR and multidrug resistance (MDR), five out of the eight *Salmonella* isolates and 33.3% of strains in total were, apart from TE, also resistant to one more antibiotic, either AMP or SXT (Figure 2a). None of the *Salmonella* strains met the criteria to be classified as an MDR strain, since non-susceptibility (i.e., resistance) to at least one agent from three antibiotic groups was not documented [37]. On the other hand, two out of the five *L. monocytogenes* isolates and 22.2% of strains in total were, apart from SXT, also highly resistant to at least CIP (Figure 2b). Moreover, one strain of *L. monocytogenes* (BF11) was identified as MDR since it demonstrated resistance to five antibiotics (CIP, MEM, P, SXT, TE) and more than three classes of antimicrobials (Figure 2b and Figure 3).

MICs for *Salmonella* strains isolated from pet food and feed, which presented resistance through AST, ranged from <0.016 to 96 μg/mL, whereas resistance was finally recorded against only one antibiotic, as seven out of the eight strains were finally deemed resistant to TE (MIC ≥ 64 μg/mL) and one strain (str. AAL 10077) was resistant to SXT (MIC = 38 μg/mL). Additionally, MIC values estimated for the highly resistant *L. monocytogenes* strains isolated from pet food and feed ranged from <0.064 to 128 μg/mL, whereas four out of the five strains (str. AAL 20148, AAL 20850, AAL 21180, BF9) were finally deemed resistant to SXT (MIC ≥ 0.5 μg/mL). The MDR *L. monocytogenes* strain BF11 ultimately demonstrated resistance against MEM, P, and TE, with recorded MIC values of 4, 4, and 48 μg/mL, respectively (Figure 3).

### 3.4. Serotyping of Salmonella spp. and L. monocytogenes Strains Isolated from Pet Food and Feed Samples

mPCR identified three major *Salmonella* serovars in pet food and feed samples; *S.* Enteritidis, *S.* Typhimurium, and *S.* Thompson. Serotyping of *L. monocytogenes* strains isolated from pet food and feed samples detected the four most common serotypes of the pathogen found in foods, namely serotypes 1/2a, 1/2b, 1/2c, and 4b, categorizing them into four distinct serogroups; PCR-serogroups IIa (ser. 1/2a, 3a), IIb (ser. 1.2b, 3b, 7), IIc (ser., 1/2c, 3c), and IVb (ser. 4b, 4d, 4e) [31].

More specifically, serotyping of *Salmonella* isolates (*n* = 15) classified them as *S.* Enteritidis (6.7%, *n* = 1), *S.* Typhimurium (6.7%, *n* = 1), and *S.* Thompson (60%, *n* = 9), while four isolates (26.7%) did not belong to the serovars identified through the experimental protocol and remained unclassified as *Salmonella* spp. (Figure 4a). For the nine *L. monocytogenes* isolates, serogroup distribution is depicted in Figure 4b and corresponds to 44.4 (*n* = 4), 11.1 (*n* = 1), 11.1 (*n* = 1), and 33.3% (*n* = 3) of the strains belonging to PCR-serogroups IIa, IIb, IIc, and IVb, respectively. The MDR *L. monocytogenes* strain BF11 belonged to serotypes 1/2b, 3b.

## 4. Discussion

In the present study, 24 microbial strains of the most commonly detected foodborne bacterial pathogens *Salmonella* spp. and *L. monocytogenes* were isolated from pet foods and animal feeding stuffs. Hence, this underscores a critical issue and point of concern that at the same time constitutes a principal component of the One Health concept [4]; pet food and feed can serve as reservoir for microbial hazards, which may subsequently end up in the food production setting and eventually the food consumer itself, either directly through petting or handling of animals and their raw feeding [38,39,40], or indirectly through human consumption of contaminated food coming from animals fed with contaminated animal feed [5,41,42]. In any case, raw pet feeding may jeopardize food safety through zoonotic transmission of the pathogens [43,44]. However, contrary to what is expected, other confounding factors such as the unattended presence and behavior of children and pets is highly more likely to contribute to the contamination and cross-contamination of food in the domestic environment, rather than the perceived inadequate hygiene practices of food consumers coming in physical contact with pet food and pets [42,45,46]. Furthermore, pet food storage and preservation in the households could play a significant role in microbial proliferation and transmission of bacterial pathogens, considering that in almost half of the cases, pet food is stored in the kitchen and some types of pet food (e.g., kibbles, treats) might be kept under elevated temperatures during their storage [40,47]. In addition, when it comes to feeding raw meat to pets, the majority of pet owners frequently reports thawing frozen meat at room temperature on the kitchen counter [48], positively contributing to the aforementioned proliferation of presumptive pathogens on the food matrix.

Despite quality assurance and the implementation of safety systems (the latter evidently following a hazard analysis critical control point (HACCP)-based approach), as well as compliance with good manufacturing and hygiene practices (GMPs/GHPs), the pet food industry faces challenges in controlling microbial hazards in animal feed [49]. Data collected from the Rapid Alert System for Food and Feed (RASFF) classify cumulatively feed materials and pet food as the second source of notifications in the EU, following only notifications related on poultry meat, whereas the most frequently notified pathogenic microorganism in the RASFF (and thus, in feed as well) was *Salmonella* [50]. A recent review of the United States (US) Food and Drug Administration (FDA) on pet food recalls in a twenty-year period (2003–2022) identified biological contamination as the primary reason for the recalls, while again, *Salmonella* serovars accounted for the majority of recall incidents followed by *L. monocytogenes* [16]. Apart from *Salmonella* spp. and *L. monocytogenes*, other bacterial pathogens, like toxigenic *E. coli* and *Clostridium botulinum*, can also be detected in pet foods [5,16]. The presence of foodborne pathogenic bacteria in pet food and animal feeding stuff can occur at any stage during the manufacturing process, storage, or even preparation of feed [51] and similar bacterial genotypes for the pathogens can be recovered and traced back in the food production chain, for example from the slaughterhouse to the animal breeding farm and its feed, highlighting specific farm management factors and practices as key points for concern [52]. Pet food is, therefore, increasingly recognized as an important vehicle for the transmission of foodborne pathogens to humans, leading to disease outbreaks of salmonellosis in most instances [53,54,55]. Consequently, there is an urged practical need for strict compliance with GMPs/GHPs to improve microbiological quality and safety of pet food and feed.

Food processing preservation methods, such as freezing and drying, may reduce the bacterial contamination in raw pet food [56,57]; nevertheless, microorganisms may still survive and pose an immediate or indirect (through cross-contamination) threat to the pet owner [49,58,59]. Moreover, the significantly reduced water activity in many low-moisture foods, including pet foods, induces cross-resistance (e.g., heat resistance) in bacterial pathogens like *Salmonella* [60,61]. Controlling this prevalent microbial pathogen and other pathogenic bacteria in dry pet foods with the use of ‘clean-label’ antimicrobials or novel non-thermal processing methods, such as high-pressure processing, seems a promising alternative to the extensive application of ‘traditional’ and chemically synthesized preservatives currently being employed [62,63,64,65].

The trend of feeding pet dogs and cats raw meat (BARF) and animal by-products because of suggested health benefits to pets as a diet more closely related to their wild-living nature has been linked with the risk of introducing antimicrobial-resistant pathogenic bacteria not only to the raw-fed pets [43,54,66,67,68,69] but also in the home food production environment [54,70]. In this study, AST results revealed notable resistance patterns among the bacterial isolates from pet food and feed. Over half of *Salmonella* isolates (53.3%) exhibited resistance to TE, while a subset of these strains (five out of eight) showed resistance to an additional antibiotic (AMP or SXT), though they remained uniformly sensitive to all other tested antimicrobial agents (AMC, CAZ, CIP, CN, CTX, FOX) (Figure 2a). Bacci et al. [67] and Fathi et al. [71] noticed similar resistance patterns to AMP and SXT. The antibiotic resistance profiles of *Salmonella* isolates obtained in this work suggest that while some frontline antibiotics commonly used to treat *Salmonella* infections (e.g., AMC and CIP) [72] remained effective, there is an emerging resistance to widely used antimicrobial agents like tetracyclines [73,74,75,76,77], which are extensively administered as pharmacological substances to treat zoonotic diseases in primary food production at the farm level [19,78,79]. Such an extensive and non-prudent use of antibiotics often leads to the occurrence of residues in food commodities [80,81]. Besides, increased AMR against SXT was observed also in more than half of *L. monocytogenes* isolates (55.6%). Implications for resistance of *L. monocytogenes* against fluoroquinolones (tested with CIP; Figure 2b) suggests that EUCAST correctly points out, in the breakpoint tables provided for the pathogen, that there is insufficient evidence that the organism is a good target for therapy with this specific antibiotic group [34]. Thankfully, all *L. monocytogenes* isolates were susceptible to E and AMP, which are the drugs of choice for treatment of listeriosis [82]. Finally, resistance of the MDR *L. monocytogenes* strain BF11 originally to five and then to three antimicrobial agents, following AST and MIC determination (Figure 2b and Figure 3), respectively, with each agent corresponding to a different antimicrobial class, further emphasizes the difficulties in antibiotic therapeutic schemes followed for treatment of listeriosis. *L. monocytogenes* strain BF11 was finally deemed resistant against penicillin, tetracyclines, and carbapenems (tested with MEM; see Figure 3). Finding MDR *L. monocytogenes* in pet food is something unusual and less commonly reported than MDR *Salmonella* [55]. In fact, this is something reported herein for the first time, to the best of our knowledge, which implies better adaptation of *L. monocytogenes* to environmental stresses and inferred development of cross-resistance, possibly assisting in the wider dissemination of the pathogen in the food chain. This finding also provides an important basis for risk early warning of drug-resistant pathogens transmitted through pet food under the “One Health” framework. Resistance of this particular *L. monocytogenes* strain (BF11) to the antibiotic group of carbapenems is alarming, since carbapenem antibiotics and MEM in particular are considered the last line of defense antimicrobial agents against AMR and MDR bacteria. Especially MEM is useful in treating *L. monocytogenes* meningitis when AMP seems ineffective or if the patient is allergic to AMP [83].

Taking together the antibiotic resistance profiles for *Salmonella* spp. and *L. monocytogenes*, the spread of antibiotic-resistant foodborne pathogenic bacteria in pet foods and feed was above 50%. In line with the reported findings by Mataragka et al. [84], the results indicate that AMR testing in animal feeding stuff is recommended to ensure antimicrobial stewardship, apart from final product sample testing. Despite the absence of molecular detection of resistance genes, the phenotypic antimicrobial resistance observed in *Salmonella* spp. and *L. monocytogenes* remains of significant clinical and public health concern. These findings highlight the potential for treatment challenges and underpin the importance of continued AMR surveillance, even in the absence of genotypic data. The latter, provided through the detection of common resistance genes (e.g., *tetA* and *tetB*), allow for a deeper understanding of bacterial gene expression and this could be the subject of future research.

Salmonellosis together with campylobacteriosis are the most reported zoonoses to humans in the EU and US [23,85]. Among the top 20 most frequently reported *Salmonella* serovars, the 2 most important and unquestionably prevalent *Salmonella* serovars associated with increased incidence of salmonellosis cases are *S.* Enteritidis and *S.* Typhimurium [23,86]. The latter along with *S.* Infantis are considered high-virulence serovars, which necessitate careful monitoring [87,88], not to mention the rapid increase in the prevalence of *S.* Infantis in poultry meat due to the presence of the pESI plasmid in a clonal dominating strain [89,90]. However, the transmission of *S.* Infantis in poultry meat through contaminated feed does not seem to be supported by our findings, since this serovar was not recovered from any poultry feed tested (Figure 4). On the other hand, phylogenetic analysis of *S.* Typhimurium is suggesting potential shared routes of transmission among human, chicken, farm, and slaughterhouse environments [91], most probably also involving animal feeding stuffs. Furthermore, in recent years, *S.* Thompson has emerged among the top 10 prevalent serotypes in China and the US [92]. As such, mPCR-based methods have been developed to differentiate between the most common clinical *Salmonella* serovars [36,93]. In our study, serotyping revealed *S.* Thompson as the predominant serovar in pet food and feed (60% of *Salmonella* spp. isolates), which correlates with the aforementioned steadily increased detection of this particular serovar [87]. While four major *Salmonella* serovars of public health relevance, namely *S.* Enteritidis, *S*. Infantis, *S.* Typhimurium, and *S.* Thompson, were screened and three of them were successfully identified in pet food and feed (Figure 4), a number of isolates (*n* = 4) remained unclassified. This represents a limitation of the applied molecular approach, as the primers used targeted a specific subset of serovars [36]. Therefore, the presence of unclassified isolates may indicate an underrepresentation of the full *Salmonella* diversity within the sampled population. Further characterization using broader or complementary typing methods would be necessary to fully elucidate the epidemiological significance of these strains.

As far as *L. monocytogenes* is concerned, the four most prevalent serotypes of the pathogen encountered in foods are 1/2a, 1/2b, 1/2c, and 4b belonging to PCR-serogroups IIa, IIb, IIc, and IVb, respectively [30,31]. The mPCR protocol of Doumith et al. [30] cannot distinguish between certain serotypes within the species *L. monocytogenes* (Table 1B), although the protocol can differentiate the four major serotypes of the pathogen from each other (serotypes 1/2a, 1/2b, 1/2c, 4b). However, serotypes 3a, 3b, 3c, 4a, 4c, 4e, 4d, and 7 are infrequently detected in food and are rarely implicated in human listeriosis cases [30], assigning in that context each one of the major serotypes of *L. monocytogenes* in the four distinct PCR-serogroups [31]. In the present study, *L. monocytogenes* PCR-serogroups IIa (44.4%) and IVb (33.3%), practically corresponding to serotypes 1/2a and 4b, were most commonly detected in pet food and feed. *L. monocytogenes* ser. 4b isolates are most frequently associated with clinical human listeriosis cases, while serotype 1/2a is mostly isolated from food products [94,95]. These serotypes are known for their role in foodborne outbreaks and, when coupled with antibiotic resistance, raise concerns about their potential impact on public health.

Present findings strenuously suggest that in the current context of One Health, the need for actions regarding policy changes and pathogen surveillance is of outmost importance. At least at the EU level, animal feed is regarded as food and is governed by the same legislation [96]. Nevertheless, there is a misconception that pet food and animal feeding stuffs may be of an inferior microbiological quality compared to the food intended for human consumption. Effective monitoring of pathogen presence and control of the microbiological quality in foods requires standardized methods. Aligned to the concept of One Health, the International Organization for Standardization (ISO) has revised from 2017 all its protocols regarding foodborne bacterial pathogen detection in foods to include the term “microbiology of the food chain” instead of the previously used term “microbiology of food and animal feeding stuffs”, so as to denote its integrated approach towards foods and feed.

As all scientific and research studies, this one has its own limitations and constraints. In terms of methodology and research process, limited access to information regarding sample size, with the non-disclosure of the total number of samples analyzed explicitly stated in this study, does not allow for further statistical processing of data, such as estimating the prevalence rates of *Salmonella* spp. and *L. monocytogenes* in pet food and feed. Lastly, the limited geographical scope of the research conducted herein, expanded only to a single country (Greece) and not a wider area or group of countries, and together with the lack of previous studies on the topic in Greece, both pose limitations in data comparisons and generally broad-term inferences regarding detection and recovery of *Salmonella* spp. and *L. monocytogenes* serotypes from pet food and feed.

## 5. Conclusions

Foodborne bacterial pathogens, like *Salmonella* spp. and *L. monocytogenes*, can be detected in the primary food production environment and feed as well. In general, pathogenic bacteria with increased AMR can be recovered from pet foods and animal feeding stuffs [67,82,83]. In strict alignment with the concept of One Health, in order to improve antimicrobial stewardship and mitigate risks stemming from inappropriate use of antibiotics in the food production chain and thus prevent potential development of MDR bacterial strains, monitoring AMR in pet food and feed is equally important as testing for AMR in clinical samples.

## Figures and Tables

**Figure 1 vetsci-12-00844-f001:**
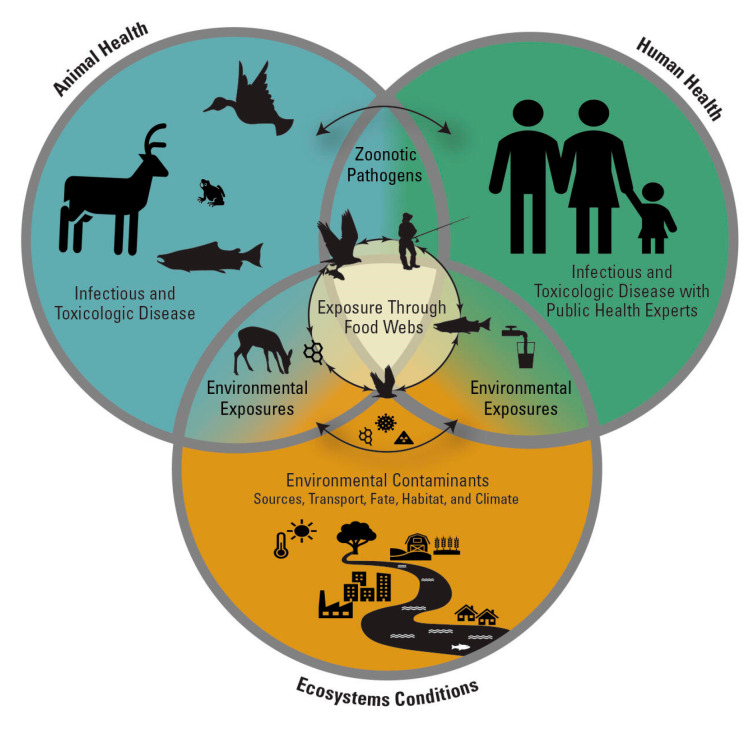
The conceptual diagram for One Health (https://www.usgs.gov/media/images/one-health-conceptual-diagram; accessed on 28 June 2025 under the creative commons license CC BY-SA 4.0).

**Figure 2 vetsci-12-00844-f002:**
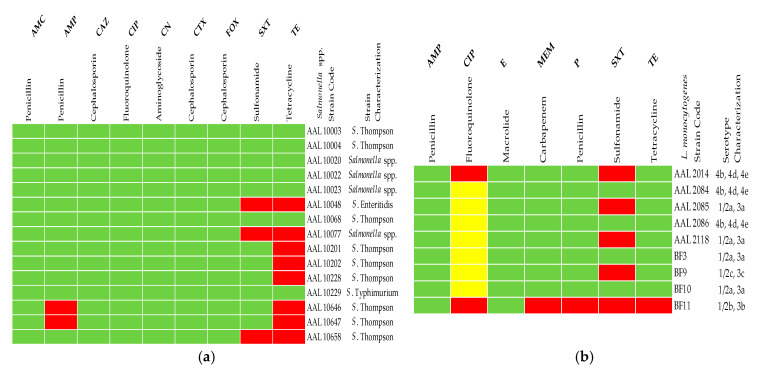
Antibiotic resistance profiles (antibiograms) of foodborne bacterial pathogens isolated from pet food and feed samples. (**a**) Antibiograms for *Salmonella* spp. isolates (*n* = 15). (**b**) Antibiograms for *L. monocytogenes* isolates (*n* = 9). The color in the antibiogram for each isolate marks its resistance pattern to the selected antibiotic; Green: Susceptible strain, Yellow: Intermediate resistant strain, Red: Resistant strain. AMC: amoxicillin-clavulanic acid, AMP: ampicillin, CAZ: ceftazidime, CIP: ciprofloxacin, CN: gentamicin, CTX: cefotaxime, E: erythromycin, FOX: cefoxitin, MEM: meropenem, P: penicillin, SXT: trimethoprim-sulfamethoxazole, TE: tetracycline.

**Figure 3 vetsci-12-00844-f003:**
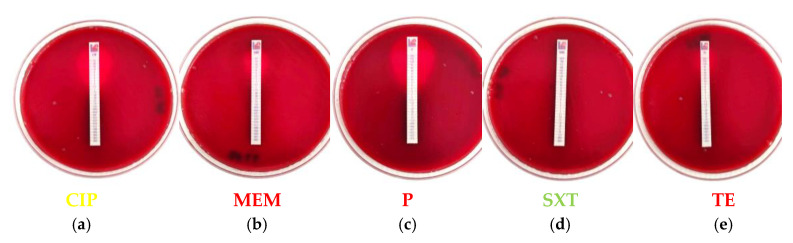
E-test for minimum inhibitory concentration (MIC) determination of the multidrug resistant *L. monocytogenes* strain BF11 isolated from raw dog food against five antimicrobial agents (antimicrobial class being designated in parenthesis). Antibiotic in green, yellow, and red ultimately marks the susceptibility, intermediate resistance, and resistance of the BF11 strain against the specific antimicrobial agent, the conferred resistance due to estimated MIC values > MIC breakpoints provided by EUCAST [33]: (**a**) MIC = 1.5 μg/mL for CIP (fluoroquinolones); (**b**) MIC = 4 μg/mL for MEM (carbapenems); (**c**) MIC = 4 μg/mL for P (penicillins), (**d**) MIC < 0.064 μg/mL for SXT (miscellaneous agents), (**e**) MIC = 48 μg/mL for TE (tetracyclines). CIP: ciprofloxacin, MEM: meropenem, P: penicillin, SXT: trimethoprim-sulfamethoxazole, TE: tetracycline, EUCAST: European Committee on Antimicrobial Susceptibility Testing.

**Figure 4 vetsci-12-00844-f004:**
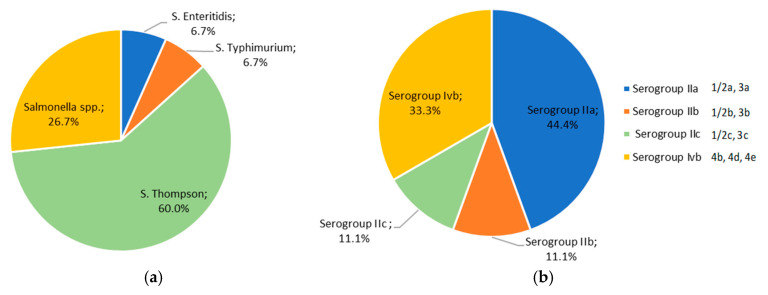
Distribution of foodborne bacterial pathogens isolated from pet food and feed samples. (**a**) Distribution of serovars for the *Salmonella* isolates. (**b**) Distribution of PCR-serogroups and serotypes for the *L. monocytogenes* isolates.

**Table 1 vetsci-12-00844-t001:** Target genomic regions and primer sequences of the mPCR used for serotyping foodborne bacterial pathogens isolated from pet food and feed samples. (**A**) Gene amplification dataset by mPCR for serotyping of *Salmonella* isolates. (**B**) Gene amplification dataset by mPCR for serotyping of *L. monocytogenes* isolates.

(**A**)			
**Target Gene**	**Sequence (5′-3′)**	**Product Size**	**Serovar Designation**
*invA*	F: GCCATGGTATGGATTTGTCC	118 bp	*Salmonella* spp.
	R: GTCACGATAAAACCGGCACT		
*sdf*	F: TGTGTTTTATCTGATGCAAGAGG	333 bp	*S*. Enteritidis
	R: CGTTCTTCTGGTACTTACGATGAC		
*fliC-r*	F: AACAACGACAGCTTATGCCG	413 bp	*S*. Infantis
	R: CCACCTGCGCCAACGCT		
*fliC-i*	F: ACTCAGGCTTCCCGTAACGC	551 bp	*S.* Typhimurium
	R: ATAGCCATTTACCAGTTCC		
*fliC-k*	F: AACGACGGTATCTCCATTGC	658 bp	*S*. Thompson
	R: CAGCCGAACTCGGTGTATTT		
(**B**)			
**Target Gene**	**Sequence (5′-3′)**	**Product Size**	**Serotype Designation**
*prs*	F: GCTGAAGAGATTGCGAAAGAAG	370 bp	*Listeria* spp.
	R: CAAAGAAACCTTGGATTTGCGG		
*ORF2819*	F: AGCAAAATGCCAAAACTCGT	471 bp	1/2b, 3b, 4b, 4d, and 4e
	R: CATCACTAAAGCCTCCCATTG		
*ORF2110*	F: AGTGGACAATTGATTGGTGAA	597 bp	4b, 4d, and 4e
	R: CATCCATCCCTTACTTTGGAC		
*lmo0737*	F: AGGGCTTCAAGGACTTACCC	691 bp	1/2a, 1/2c, 3a, and 3c
	R: ACGATTTCTGCTTGCCATTC		
*lmo1118*	F: AGGGGTCTTAAATCCTGGAA	906 bp	1/2c and 3c
	R: CGGCTTGTTCGGCATACTTA		

F: Forward primer; R: Reverse primer.

**Table 2 vetsci-12-00844-t002:** Antimicrobial susceptibility testing (AST) of foodborne bacterial pathogens isolated from pet food and feed samples. (**A**) AST of *Salmonella* isolates (*n* = 15) per antibiotic tested (% of total). (**B**) AST of *L. monocytogenes* isolates (*n* = 9) per antibiotic tested (% of total).

(**A**)											
**Strain**	**AMC (%)**	**AMP (%)**	**CAZ (%)**	**CIP (%)**	**CN (%)**		**CTX (%)**	**FOX (%)**	**SXT (%)**		**TE (%)**
Resistant	0 (0.0)	2 (13.3)	0 (0.0)	0 (0.0)	0 (0.0)		0 (0.0)	0 (0.0)	3 (20.0)		8 (53.3)
Intermediate	0 (0.0)	0 (0.0)	0 (0.0)	0 (0.0)	0 (0.0)		0 (0.0)	0 (0.0)	0 (0.0)		0 (0.0)
Susceptible	15 (100)	13 (86.7)	15 (100)	15 (100)	15 (100)		15 (100)	15 (100)	12 (80.0)		7 (46.7)
Total	15	15	15	15	15		15	15	15		15
(**B**)											
**Strain**	**AMP (%)**	**CIP (%)**	**E (%)**	**MEM (%)**		**P (%)**		**SXT (%)**		**TE (%)**	
Resistant	0 (0.0)	2 (22.2)	0 (0.0)	1 (11.1)		1 (11.1)		5 (55.6)		1 (11.1)	
Intermediate	0 (0.0)	7 (77.8)	0 (0.0)	0 (0.0)		0 (0.0)		0 (0.0)		0 (0.0)	
Susceptible	9 (100)	0 (0.0)	9 (100)	8 (88.9)		8 (88.9)		4 (44.4)		8 (88.9)	
Total	9	9	9	9		9		9		9	

AMC: amoxicillin-clavulanate, AMP: ampicillin, CAZ: ceftazidime, CIP: ciprofloxacin, CN: gentamicin, CTX: cefotaxime, E: erythromycin, FOX: cefoxitin, MEM: meropenem, P: penicillin, SXT: trimethoprim-sulfamethoxazole, TE: tetracycline.

## Data Availability

Restrictions apply to the availability of data. Data on the number of tested samples (pet food and feed) are not readily available because those data are confidential. Requests to access this data could be directed to Eurofins Athens Analysis Laboratories and the Hellenic Army Biological Research Centre. All other contributions and datasets deriving from the original data which are presented in the study are included in the article/Appendix A, and any further inquiries can be directed to the corresponding author.

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
