# Peer review of "Serotyping and Antibiotic Resistance Profiles of Salmonella spp. and Listeria monocytogenes Strains Isolated from Pet Food and Feed Samples: A One Health Perspectiveâ€"

_vetsci, 2025, doi:10.3390/vetsci12090844_

Round 1
Reviewer 1 Report
Comments and Suggestions for Authors
I thank the Authors for their work. I reported my suggestions in the attached file.

The quality of English language is good, I had some doubts about certain expressions.
Reviewer 2 Report
Comments and Suggestions for Authors
This manuscript describe serotyping and antibiotic resistance profiles of 2 strains isolated from raw pet food. Topic is very interesting, but I have some suggestions for authors:
Title: As I understand you analized only BARF and raw chicken, so I suggest to modify „pet food and feed samples” in the title of manuscript. I’m also not sure if one health perspective is a key aspect in this work
Lines 66-70 – You should rearrange this sentence because it's very long, has a lot of insertions, and is difficult to understand. I suggest breaking it up into shorter sentences.
Figure 1 is completely illegible. You should enlarge it.
Introduction: The introduction extensively discusses "One Health," but key information regarding the research conducted is missing. What have been the previous results of microbial testing of raw and commercial pet foods? Have there been any comercial pet food recalls due to the pathogens tested? The studies focus on raw food and chicken. Homemade raw food is typically made from meat intended for human consumption. Therefore, people eat the exact same meat. Therefore, since we eat the same meat anyway, does making pet food from the same meat increase the risk of human infection? This is meat we buy anyway, so how does this relate to the "One Health" concept?
Lines 95-108 are a little bit chaotic, they should be rewritten.
109-114 - The hypothesis should be corrected.
Unit 2.1 - How many samples were there? Was BARF purchased from a company or home-made? What was the composition of the BARF? Where did the tested chicken come from? What was the source of the meat?
Figure 2: What are these numbers?
Results: How many samples were tested and in how many of them were these strains found and in how many were they not found?
Line 271 – again, If the meat was used for human consumption, we still come into contact with it
Line 272 - Are you suggesting that humans consume pet food?
Lines 278-282 - Now you mention dry food and treats, but you have investigated raw food that is not stored at an elevated temperature.
Lines 282-285 - Some people undoubtedly defrost BARF and meat on the kitchen counter. But is it kept there long enough after thawing to promote the growth of pathogens? Or is it served immediately after thawing?
Lines 265-316 - This section is not a discussion of the results. This section, in a concise form, should be included in the introduction.
Lines 294 – 299 - These data refer to processed, not raw, foods. This suggests that heat treatment and processing do not eliminate pathogens from the products. The study focuses on pathogens in raw foods, not processed foods.
Lines 299-306 - The results are for raw food, but processed food is also discussed here, which may be confusing to the reader.
Line 307 – so, freezing is the method using in raw food, you should emphasize that this is the method which is used in your material – BARF is usually portioned and frozen
Lines 312-316 – again you mentioned dry food, this is not the subject of this work
Lines 317-321 – and again, if we use the same meat that we eat, so do we introduce “new” pathogens using this meat for feeding pets? Pathogens are often found in commercial, highly processed pet foods, suggesting that by using them, we introduce additional pathogens into our home environment. So, by purchasing the same meat we eat (pathogens we already have at home from eating meat), are we introducing new pathogens into our home?
Do cats and dogs have any adaptations to cope with things like Salmonella? Using cats as an example, consider the shorter digestive tract in cats, the lower pH in the stomach, and the evolutionary adaptation to eating raw meat and dealing with bacteria.
I suggest you describe the results from a different angle and resubmit the manuscript for review.
Reviewer 3 Report
Comments and Suggestions for Authors
Essential Revisions:
- Expand Discussion: Link findings to One Health actions (e.g., policy changes, surveillance).
- Clarify Methods: Detail sampling strategy and statistical approaches.
- Improve Data Presentation: Revise figures/tables for clarity; integrate supplementary data.
- Update References: Add recent literature on pet food AMR and serotype epidemiology.

Reviewer 4 Report
Comments and Suggestions for Authors
This study systematically reveals for the first time the contamination status of Salmonella and Listeria (types 1/2a and 4b) in pet food and feed in Greece and their high resistance rates to key antibiotics such as tetracycline and co-trimoxazole, and finds a Listeria strain carrying a multidrug resistance phenotype, which provides an important basis for risk early warning of drug-resistant pathogens transmitted through pet food under the framework of "One Health". However, I have minor concerns:
(1) The description of sample source, collection batch and geographical distribution is insufficient, which affects the extrapolation of results; it is recommended to supplement the sampling plan, batch and regional distribution table, and increase the representative description of samples;
(2) The untyped strains are only taken by "Salmonella spp.", and there is a lack of follow-up identification ideas; if possible, please use whole genome sequencing or supplementary PCR protocols for untyped Salmonella to clarify their taxonomic status; if can’t please add limitations of this study.
(3) Methods: 2.2 section, please Clarify the EUCAST version (e.g. 2025 version) and drug resistance determination criteria
(4) No molecular detection of drug resistance genes, it is difficult to distinguish acquired drug resistance from intrinsic drug resistance; it is recommended to use PCR or WGS to detect common drug resistance genes (such as tetA/B,blaTEM, sul, etc.). please Deepen the discussion of the clinical significance and limitations of this study
Round 2
Reviewer 1 Report
Comments and Suggestions for Authors
I thank the Authors for their comments and clarifications.
Author Response
We would like to thank the Reviewer for accepting our responses to the comments and suggestions made during the peer-review process.
Reviewer 2 Report
Comments and Suggestions for Authors
You can find my comments and suggestions in the attachement.
